# The Curled Up Dimension in Quasicrystals

Fang Fang [1,*] , Richard Clawson [1,2] and Klee Irwin [1]

1 Quantum Gravity Research, Los Angeles, CA 90290, USA; Richard@QuantumgravityResearch.org (R.C.); Klee@QuantumGravityResearch.org (K.I.)

2 Faculty of Health, Engineering and Sciences, University of Southern Queensland, Toowoomba, QLD 4350, Australia

* Correspondence: Fang@QuantumGravityResearch.org; Tel.: +1-310-574-6934

**Abstract:** Most quasicrystals can be generated by the cut-and-project method from higher dimensional parent lattices. In doing so they lose the periodic order their parent lattice possess, replaced with aperiodic order, due to the irrationality of the projection. However, perfect periodic order is discovered in the perpendicular space when gluing the cut window boundaries together to form a curved loop. In the case of a 1D quasicrystal projected from a 2D lattice, the irrationally sloped cut region is bounded by two parallel lines. When it is extrinsically curved into a cylinder, a line defect is found on the cylinder. Resolving this geometrical frustration removes the line defect to preserve helical paths on the cylinder. The degree of frustration is determined by the thickness of the cut window or the selected pitch of the helical paths. The frustration can be resolved by applying a shear strain to the cut-region before curving into a cylinder. This demonstrates that resolving the geometrical frustration of a topological change to a cut window can lead to preserved periodic order.

**Keywords:** quasicrystals; aperiodic order; periodic order; perpendicular space; geometric frustration; curled-up dimensions

## 1. Introduction

With the announcement by Shechtman et al. [1] in 1984 of a physical material exhibiting crystallographically forbidden symmetries in its diffraction patterns, and the following theoretical description by Levine et al. [2], the study of aperiodically ordered structures (dubbed quasicrystals) moved from what was something of a niche specialty to a burgeoning field in both mathematics and material science. Like crystals, quasicrystals have long range order, usually the symmetry of some regular polyhedron, and they have sharp diffraction patterns, generally because they are built up from a finite number of primitive structural units. But unlike crystals, these structural units are not periodically repeated, so quasicrystals lack symmetry under any translation group. This allows for rotational symmetries incompatible with periodicity, e.g. icosahedral, while undermining the main theoretical tools (such as the Bloch theorem and standard Brillouin zones) that are used to understand crystalline materials [3–5].

In time, theory was adapted to the new structures and great progress was made in understanding their properties, in particular, with hydrodynamic [6] and effective field theory [7] models. The foundation of this, of course, is having tractable models of their basic geometry, and a key to this work was the cut-and-project description (also known variously as the superspace, or model set, description) where points of a quasicrystal are identified with projections of points belonging to a periodic lattice in a higher dimensional space [3,8].

In other work, the rotational symmetry's incompatibility with periodicity is identified as a type of geometric frustration. This may be relieved locally by curving into a higher dimension [9,10], or globally with the idea of distortion (nonmetricity) and disclination [11] when focusing only on the quasicrystal space. In all these cases, the quasicrystal, which

lacks translational periodicity itself, is seen as a distortion of some periodic structure, and this connection provides a useful tool for modeling and analysis. This paper introduces a method to see directly the periodicity in the quasicrystal by looking at its dual space, the perpendicular space of the cut-and-project description. The periodicity in the perpendicular space not only gives a straightforward solution for the geometric frustration in the quasicrystal space, but also defines a new kind of "closeness" for the point set in the quasicrystal.

Section 2 reviews the cut-and-project method of generating a quasicrystal and the duality between the quasicrystal and its perpendicular dual space. They both appear aperiodic if looked at individually. Section 3 points out that when looking at the perpendicular space based on the sequential order of the quasicrystal points, if one glues the opposite boundaries of the perpendicular space together, the order appears to be perfectly periodic. The reverse can also be said, that the periodic order can be revealed in the quasicrystal if looking at it according to the order of its perpendicular space. Section 4 suggests a solution to the geometric frustration in the the $P_\perp + P_\parallel$ space, by curving the perpendicular space into a loop, and adding shear to align the frustrated boundary. Section 5 briefly summarizes and suggests continued research.

## 2. Quasicrystals and the Perpendicular Space in the Cut-and-Project Approach

The method of cut-and-project from a higher-dimensional parent lattice is one of the main tools for generating and studying quasicrystals [3,12,13]. Indeed, while other methods such as inflation rules, matching rules, or generalized dual multigrids are often used, in nearly all cases the quasicrystal also admits of a cut-and-project construction (see for example [3–5]). One might even go so far as to say "all" rather than "nearly all" quasicrystals can be so constructed, but the term "quasicrystal" is used in a variety of contexts and there does not appear to be consensus on a universal formal definition whereby one could make such a claim rigorous. (Exceptions to the claim are suggested by Baake and Grimm's discussion of aperiodic order beyond cut-and-project sets (or model sets), such as the Thue–Morse and Rudin–Shapiro chains Chapter 10 of the [14], and by Burdik et al's study of quasicrystals based on so-called $\beta$-integers [15].) Nevertheless, the cut-and-project method applies to at least a very general class of quasicrystals, which are the subject of this note.

To illustrate the cut-and-project, we take a one dimensional quasicrystal, the Fibonacci chain, as an example. Start with the $\mathbb{Z}_2$ lattice (Figure 1a). The 1D quasicrystal space, $P_\parallel$, is at an angle $\theta = \arctan(\Phi)$, where $\Phi$ is the golden ratio, to the root vectors of the $\mathbb{Z}_2$ lattice. Its perpendicular space $P_\perp$, by definition perpendicular to $P_\parallel$, contains a finite segment called the cut window. The cut window is extruded along the direction of $P_\parallel$ to form the cut region, an infinite region of fixed width which captures all the points in the $\mathbb{Z}_2$ lattice that project to $P_\parallel$ as the point set of the Fibonacci chain (points on solid line in Figure 2a). Note, incidentally, that one obtains the same Fibonacci chain if one sets the positive $P_\parallel$ at the angle $\theta$ with the $-l_1$ lattice vector, so that it slopes up to the right in Figure 1a instead of down to the right.

Although the points in the cut region extend without limit in the infinite $P_\parallel$, they are bounded in $P_\perp$ within the finite span of the cut window, as mentioned above. Therefore, the point set in the cut window is dense. There is a dual nature between the points' projections in these two spaces. Points that are closer in the quasicrystal space tend to space away from each other in the perpendicular space, and vice-versa. More importantly, there is a pattern to the spacing.

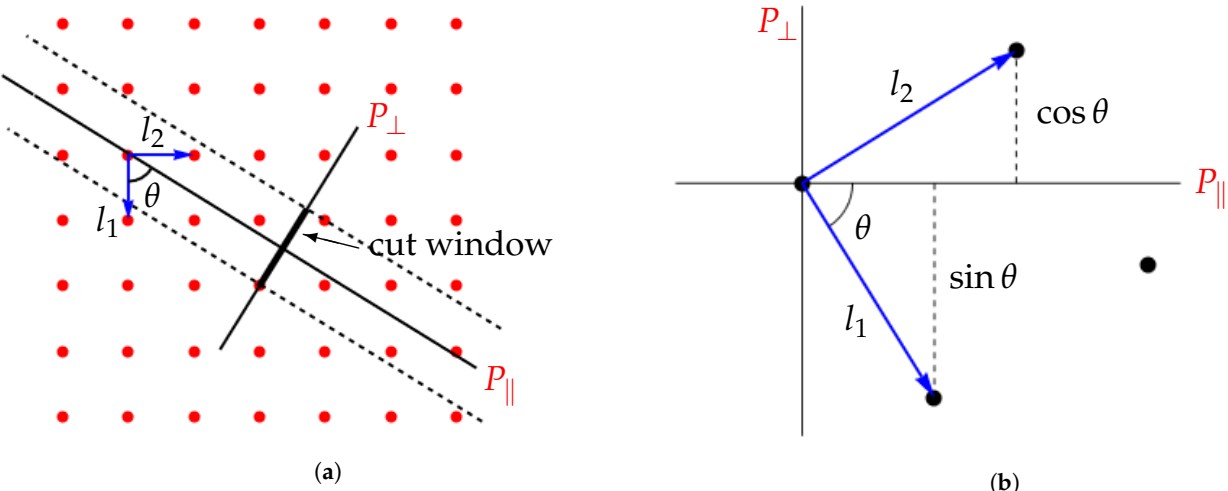

(**a**)                          (**b**)

**Figure 1.** Cut-and-project scheme for Fibonacci chain. (**a**) $\mathbb{Z}_2$ parent lattice with lattice vectors $l_1$ and $l_2$, showing $P_\perp$, $P_\parallel$, and cut region bounded by dashed lines. $\tan\theta = \Phi$, golden ratio. (**b**) Detail of parent lattice (rotated so $P_\parallel$ is horizontal) showing distances between points as projected in $P_\perp$.

## 3. Periodic Order in the Curled Up Perpendicular Space

The $P_\perp$ distance between points is shown in Figure 1b, with $\cos\theta$ and $\sin\theta$ as the short and long lengths, respectively. Following a sequence of points from left to right in the quasicrystal space as shown in Figure 2a, and comparing with Figure 1b, we see that when a position increment in $P_\perp$ is positive, the displacement is $\cos\theta$, while if the increment is negative, it is $-\sin\theta$.

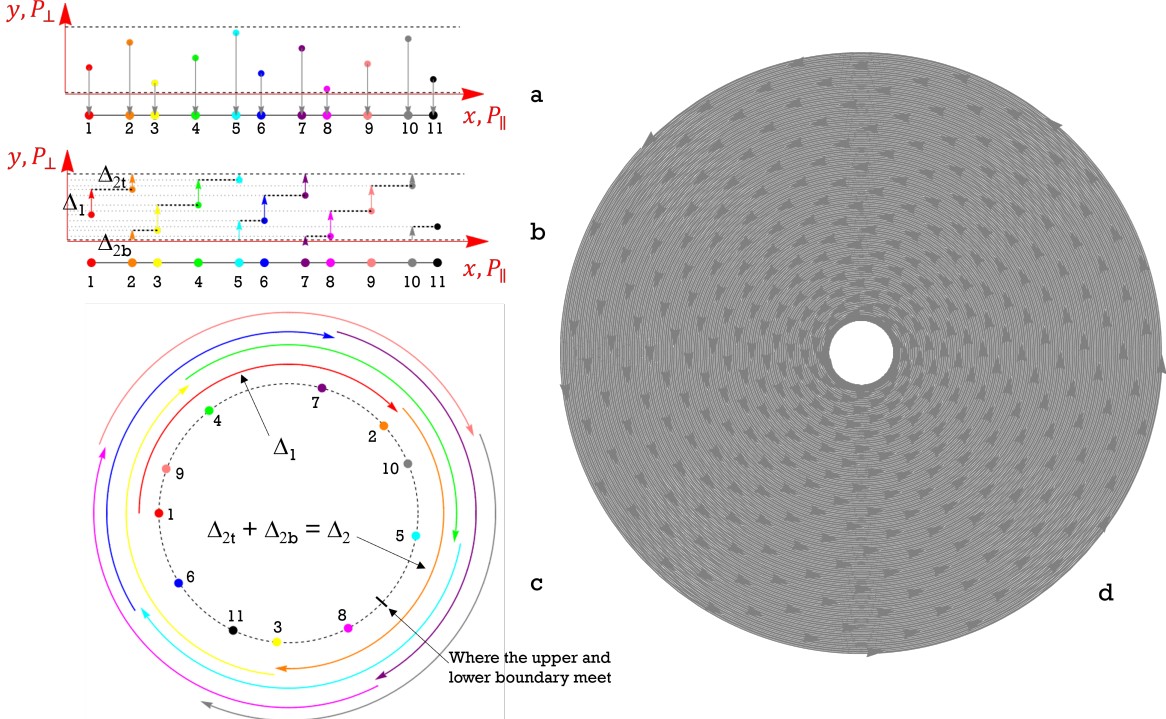

**Figure 2.** (**a**) Cut region in $\mathbb{Z}_2$ parent lattice (region bounded by top and bottom dashed lines) and projection in $P_\parallel$ (on solid line). (**b**) Periodicity after identifying top and bottom boundaries of $P_\perp$, showing equal increments in $P_\perp$ (examples $\Delta_1$, $\Delta_{2b}$, and $\Delta_{2t}$ labeled). (**c**) Same periodicity illustrated after wrapping $P_\perp$ into a loop. Dashed circle is $P_\perp$; colors match points and increments in (**a**,**b**) with points and arcs in (**c**). (**d**) Displaying the successive arc segments and arrow tips with increasing radius spreads the circle into an annulus, making the self-similarity of the structure more visually apparent.

Now, if we consider the cut window as a loop (gluing the bottom and top boundary together), we can take all displacements as positive, as shown by the colored vertical arrows in Figure 2b. Formally, with $y$ as the $P_\perp$ coordinate and $\{y_t, y_b\}$ as the top and bottom cut window boundaries, we count the perpendicular space interval between successive points as

$$\Delta_n = \begin{cases} y_{n+1} - y_n & y_{n+1} > y_n \\ \Delta_{nt} + \Delta_{(n+1)b} & y_{n+1} < y_n \end{cases} \tag{1}$$

$$\Delta_{nt} = y_t - y_n, \quad \Delta_{(n+1)b} = y_{(n+1)} - y_b. \tag{2}$$

This amounts to using a sort of periodic coordinate for $P_\perp$. When the displacement is undivided, it is a short segment of length $\cos\theta$, as seen in Figure 1b. When the displacement is divided by crossing the cut window boundary, a sum of two terms occurs. This sum is the complement, within the cut window, of the long segment $\sin\theta$. The total length of the cut window is the $P_\perp$ projection of a unit cell, $\cos\theta + \sin\theta$, so this complement has length $(\sin\theta + \cos\theta) - \sin\theta = \cos\theta$. Thus, the positive displacement in periodic coordinates is always $\cos\theta$ for both divided and undivided segments, $\Delta_1 = \Delta_{2t} + \Delta_{2b} = \Delta_3 = \Delta_4 = \Delta_{5t} + \Delta_{5b} = \ldots$. As a fraction of the cut window, this increment is

$$\frac{\Delta_n}{|\text{cut window}|} = \frac{\cos\theta}{\cos\theta + \sin\theta} = \frac{1}{1 + \tan\theta} = \frac{1}{1 + \Phi} = \frac{1}{\Phi^2}. \tag{3}$$

When the cut window is curved into a loop, the angular distance between these successive points is therefore always the same, $2\pi/\Phi^2$, a periodicity incommensurate with the full circle. This is shown in Figure 2c for the first few points; repeated for many points, it becomes like a phyllotaxis pattern as in Figure 2d. (One should remember, however, that the disk-like nature of this representation is just for illustrative purposes, to exhibit the structure of the pattern; the actual perpendicular space, when curved into a loop, is just a circle, not a disk or annulus.) Incidentally, if one uses the alternate construction mentioned for the Fibonacci chain, making $P_\parallel$ slope up to the right in Figure 1a, then Figure 1b becomes inverted, with $\sin\theta$ above and $\cos\theta$ below. In that case, the positive increment $\Delta_n$ would be the larger $\sin\theta$ instead of the smaller $\cos\theta$, and Equation (3) would yield $1/\Phi$ instead of $1/\Phi^2$. The points on the circle would still make the same pattern, but in the reverse direction, because an angle on the circle of $+2\pi/\Phi$ is equivalent to an angle of $-2\pi/\Phi^2$, since $\Phi^{-1} + \Phi^{-2} = 1$.

Another interesting pattern, in addition to this periodic angular advancement, is that every new added point lands in the longest empty section of the circle to maximize the uniformity of distribution of the points on the circle. In a way, the adjacency in the quasicrystal space results in a periodic repulsion in $P_\perp$, or, alternatively, a new "closeness" can be defined in the looped $P_\perp$ in terms of the periodicity. Specifically, we may define a "periodic distance" in $P_\perp$ as the number of integral $\frac{2\pi}{\Phi^2}$ periods separating points. Then, points that are periodically close in $P_\perp$ are physically close in $P_\parallel$, and vice versa. This contrasts with the ordinary Euclidean closeness in $P_\perp$ which induces a sort of "repulsion" in $P_\parallel$ (resembling the same-sign charge distribution), and vice versa.

This periodicity in $P_\perp$ can be extended to many higher dimensional quasicrystals, where $P_\perp$ is also higher dimensional. In this case, the cut window is often a polytope and the cut region is the prism formed by extruding that polytope in the directions of $P_\parallel$. The cut window usually has matching opposite faces (i.e., congruent and parallel). (Exceptional cases include cut windows with fractal boundaries [14,16], to which this analysis would not be apply.) By focusing on the one-dimensional subspace normal (within $P_\perp$) to a pair of opposite faces, we can see the same periodicity as described above. This corresponds to the quasiperiodic structure along a one-dimensional line in the quasicrystal space. Perhaps it is even possible to wrap the entire cut window polytope into a closed manifold by identifying each face with its opposite, just as we wrapped the 1D cut window of the Fibonacci chain into a circle.

Returning to our 1D case, the distribution of points in $P_\perp$ also has a self-similar nature, which resembles the self-similar nature of the P-adic numbers [17]. This is in fact the self-similarity of the inflation/deflation structure of Fibonacci chains under substitution rules. It can be seen on the circle of $P_\perp$ as illustrated in the sequence of Figure 3. With just the first two points, the circle is divided into two unequal arcs $L$ and $S$ whose length ratio is golden. As mentioned above, the next point lands on the largest gap, the single $L$ arc. It divides that into two arcs related by the golden ratio, so that the longer of them is the same arc length as the $S$ arc that was not divided. This yields a circle with three arcs of only two distinct lengths, again in the golden ratio, where the previous $S$ is the new $L$. Since there are now two $L$ arcs, adding the next two points will divide each of the $L$'s, and again we find the entire circle divided into arcs of only two distinct lengths whose ratio is golden, the previous $S$ being the new $L$. With each successive step, where the number of points added in a step is the number of $L$ arcs on the circle, we get an inflation with $S \to L$ and $L \to LS$. This proceeds without limit, yielding an endless (approximate) self-similarity of the Fibonacci structure of the points as projected onto the cut window, which is closed upon itself into a circle. By displaying successive points on circles of increasing radii (Figure 2d) one spreads the circle into a thick annulus, and the self-similar structure is made more visually striking.

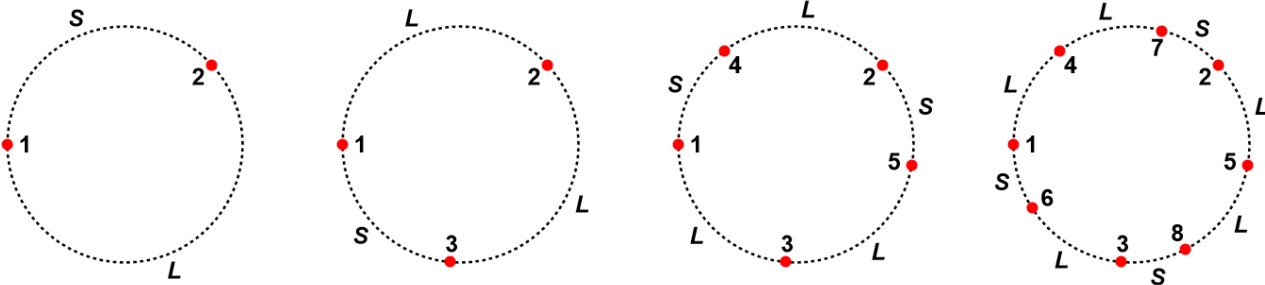

**Figure 3.** Self similarity of point distribution on the $P_\perp$ loop. In a given step, there are $n$ long arcs ($L$) of angle $\frac{2\pi}{\Phi^n}$, short arcs ($S$) having angle $\frac{2\pi}{\Phi^{n+1}}$. In next step, $n$ new points are added by incrementing the angle in steps of $\frac{2\pi}{\Phi^2}$. These all land in $L$ arcs, dividing them into $LS$ arcs at new scale, with new $L$ having angle $\frac{2\pi}{\Phi^{n+1}}$. The number of $L$ arcs in successive steps are Fibonnaci numbers 1, 2, 3, 5. . . .

The self-similarity is not exact because the initial point set cannot be partitioned into subsets that all reproduce the original at a fixed scale, but rather into two groups of subsets that reproduce it at two different scales. Specifically, at any level of inflation, the Fibonnaci word can be partitioned into a set of $L$'s and $S$'s. Every $L$, when filled in by inflation, reproduces the pattern of the original circle at a scale reduced by some $\Phi^n$, while each $S$ when filled in reproduces the original pattern at a scale reduced by $\Phi^{n+1}$. The concept of multifractal analysis suggests that we calculate the similarity dimension of each of these subgroups separately. (We refer to Strogatz Chapter 11 of the [18] for an elementary introduction to the analysis of fractal dimension).

For a number of partitions $N(r)$ at scale factor $r$, the similarity dimension is

$$D = \frac{\ln N(r)}{\ln r}. \tag{4}$$

In the Fibonacci chain at the $n$th level, the number of $L$'s (reduction factor $\Phi^n$) is the Fibonacci number $F_n$, while the number of $S$'s (reduction factor $\Phi^{n+1}$) is $F_{n-1}$. Then, using Binet's formula, $F_n = (\Phi^n - \Phi^{-n})/\sqrt{5}$, the respective similarity dimensions for these two groups are

$$D_L = \frac{\ln F_n}{\ln \Phi^n} = \frac{\ln(\Phi^n - \Phi^{-n}) - \ln \sqrt{5}}{\ln \Phi^n} \qquad \rightarrow 1 \quad \text{for large } n$$

$$D_S = \frac{\ln F_{n-1}}{\ln \Phi^{n+1}} = \frac{\ln(\Phi^{n-1} - \Phi^{-(n-1)}) - \ln \sqrt{5}}{\ln \Phi^{n+1}} \qquad \rightarrow 1 \quad \text{for large } n.$$

(5)

This seems a fairly trivial result for the fractal dimension. It is a reflection of the fact that unlike, e.g., the Cantor set, for any level of inflation we find that each interval filled by a copy of the set is in turn completely filled by the reduced copies at their respective scales. Among other things this means that the full point set is dense on the circle.

A calculation of the Minkowski dimension also yields unity, if the boxes are taken to cover either the long or short arcs. For box size $\epsilon = \Phi^{-n}$ the number of boxes is $N(\epsilon) = F_n$, and the Minkowski dimension $\ln N(\epsilon) / \ln(\epsilon^{-1})$ gives the same expressions as $D_L$ and $D_S$ above. This reflects the fact that unlike e.g. the Koch curve, the points of the Fibonacci inflations all lie on the circle or, effectively, on a finite interval in $\mathbb{R}$ (where, as mentioned before, they are dense). The correlation dimension [19] and a numerical calculation of the pointwise dimension also give the same results. We thus have a set with an inexact, nontrivial, rigorously structured type of self-similarity, which neither leaves finite gaps in the topological space it lives in, nor "spreads" itself in the usual fractal sense into a larger topological space. Consequently, its fractal dimension is trivial, in the sense that it matches the topological dimension.

### 4. "Frustration" in the $P_\perp + P_\parallel$ Space and Shear as a Solution

Geometric frustration refers in general to the conflict between some nontrivial local order and the global environmental order, or as described by Sadoc et al. [9], the condition that local order cannot propagate "freely" throughout the space. An example is the local maximum sphere packing arrangement in a global maximum arrangement, or the breaking of a periodic pattern due to some incompatible local order. Geometric frustration is a common phenomenon in quasicrystals due to their aperiodic nature [10]. For example, the icosahedral symmetry of many 3D quasicrystals is not compatible with a crystalline propagation to fill flat space, but the frustration can be resolved by curving the space into the fourth dimension and propagating the order on a 3-sphere in 4D.

In curling up a dimension in $P_\perp$, we find a new kind of geometric frustration, as well as a way to resolve it. To see this, we start by looking at the "parent" space, the projection space that is the sum of $P_\parallel$ and $P_\perp$. The 1D Fibonacci chain is the projected image in $P_\parallel$ of points in the cut region of the 2D parent space, as shown in Figure 4. Firstly, we identify a linear trajectory in the cut region connecting points that are adjacent in $P_\parallel$ (gray segments shown in Figure 4(a1)). As discussed in the previous section, the step advancement in $P_\perp$ is periodic. In $P_\parallel$, however, there is a "frustration" for every cycle in $P_\perp$, as shown Figure 4(a1) by the intervals labeled **d**. This frustration breaks the periodicity in $P_\parallel$.

As we curve $P_\perp$ into a circle leaving $P_\parallel$ invariant, the cut window trajectory becomes helical, with the linear axial component as the propagation in $P_\parallel$ and the angular (phase, clock) component as the propagation in $P_\perp$. The frustration manifests as a line defect shown in Figure 4(b1), where the helical path breaks at the line where the top and bottom boundaries of $P_\perp$ meet. The breaks can be eliminated and the trajectory segments aligned into a continuous helix by shearing the cut region along the axial direction so that a rectangle is deformed into a parallelogram (Figure 4(a2)), thereby vertically aligning the endpoints of the trajectory segments before curving the $P_\perp$ into a cylinder (Figure 4(b2)).

Different choices of discrete shear strain are possible to join one section with the others. A shear that connects a section with its first, second, third etc., neighbors on either side, respectively, creates a single, double, triple helix, etc. (Figure 5). The shear may even create isolated loops, if it connects each section back to itself. For cut windows of different sizes, the minimum shear needed to produce a single helix increases as the cut window gets thicker (Figure 6).

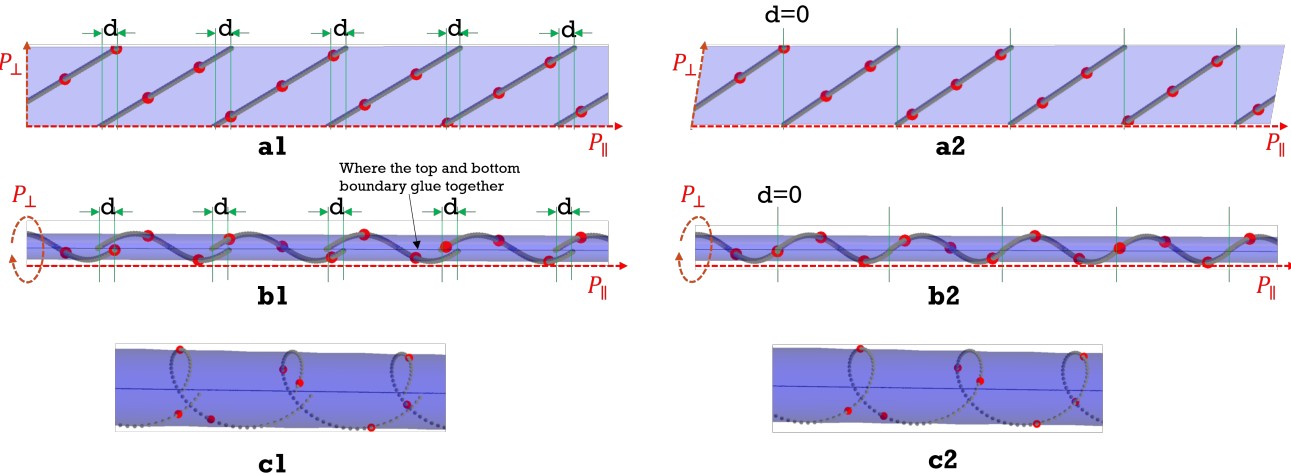

**Figure 4.** Cut region (**a1**) wrapped into a cylinder (**b1**), with an exaggerated view to better see mismatch (**c1**). When introducing shear, cut region (**a2**) becomes a parallelogram and mismatches in cylinder close to form a true helix (**b2**, **c2**).

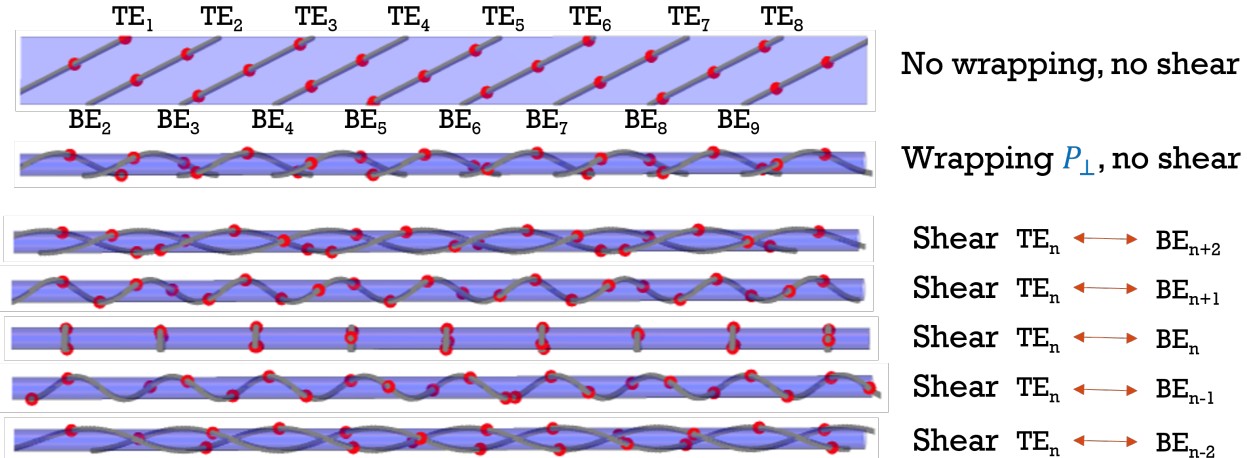

**Figure 5.** When the cut region is curved into a cylinder, different choices of shear connect different sections to make different helices. $n$ helices are created when the shear connects each sections with its $n^{\text{th}}$ neighbors, rather than just its immediate neighbors.

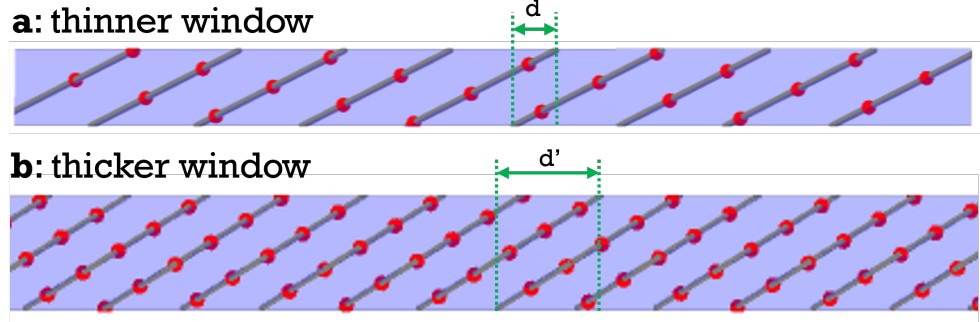

**Figure 6.** Comparing cut windows of different thickness: (**a**) thinner window requires a smaller shear correction $d$ to connect a section with its first neighbors and make a single helix; (**b**) thicker window requires a larger shear correction $d'$ for a single helix.

## 5. Summary and Discussion

In an attempt to resolve the geometric frustration in quasicrystals, which is a natural result of their aperiodic nature, we discovered the inherent periodicity associated to a quasicrystal when considering its dual space, the perpendicular space ($P_\perp$) of the cut-and-project method. The periodicity in $P_\perp$ can be better seen when curving $P_\perp$ and gluing its upper and lower boundaries together. Using the $\mathbb{Z}_2$ lattice projecting to a 1D Fibonacci chain (in $P_\parallel$) as an example, $P_\perp$ is a segment, which after being curved becomes a loop. The periodicity of the points, incommensurate with the period of the $P_\perp$ loop, fills the loop with a dense point set that exhibits a 2-scale self similarity, but the fractal dimension of this set is just unity, matching the topological dimension of the loop itself.

After putting $P_\parallel$ and the curved $P_\perp$ together, the geometric frustration, or the disjoints in the $\mathbb{Z}_2$ to Fibonacci chain case, are clearly shown in this cylindrical space. A shear needs to be introduced to connect the disjoints, or resolve the geometric frustration. Different degrees of shear can result in different numbers of helices in the cylindrical space. The concept generalizes naturally to higher-dimensions when the cut window is a polytope with parallel opposing facets.

This paper also discussed a new kind of "closeness" based on the periodic nature of $P_\perp$. Points that are ordinarily close in $P_\perp$ are far apart in $P_\parallel$, but points that are "periodically" close in $P_\perp$ (separated by few integral periods) are also close in $P_\parallel$.

The author will expand the research in higher dimensional quasicrystals and also study the curved $P_\perp$ as the phase space (in complex plane) of the quasicrystal to understand deeper the interplay between $P_\perp$ and $P_\parallel$ and the connection between periodicity and aperiodicity.

**Author Contributions:** F.F. discovered the periodicity in the perpendicular space of quasicrystals; F.F. and R.C. identified its connection to the frustration in the curled-up cut region; K.I. led the overall project in research on geometric frustration in quasicrystals. F.F. and R.C. wrote the paper. All authors have read and agreed to the published version of the manuscript.

**Funding:** This research received no external funding.

**Acknowledgments:** The authors thank Dugan Hammock, David Chester, and Raymond Aschheim for their help in optimizing the code and fruitful discussions. We also thank the anonymous reviewers for many helpful comments.

**Conflicts of Interest:** The authors declare no conflict of interest.

## Abbreviations

The following abbreviations are used in this manuscript:

MDPI　　Multidisciplinary Digital Publishing Institute
DOAJ　　Directory of open access journals

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
