# Peer review of "The Curled Up Dimension in Quasicrystals"

_crystals, doi:10.3390/cryst11101238_

Round 1

Reviewer 1 Report

The authors discuss an interesting way of finding hidden periodicity in quasicrystals aperiodic structures by looking at the (non-physical) transverse space and its compactification. The manuscript does not contain many results but it is well written and to the point. The relevance of the result is not clear from the physical point of view but it is definitely interesting from a mathematical one and it might be potentially useful to understand better the geometric structure of quasicrystals.

Here are my minor concerns to improve the manuscript:

(1) The authors explain that MOST of quasicrystals can be generated via cut-and-project method. It would be nice to add an explanation or at least an example of a case which cannot be generated in that way and briefly clarify why.

(2) The manuscript should be a bit more complete with respect to the previous literature. For example, in the first sentence let us add at least a few introductory works on the dynamic and geometry of quasicrystals. 

My suggestions: 

Quasicrystals: A Primer - C. Janot - Oxford University Press (oup.com)

SciPost: SciPost Phys. 9, 062 (2020) - Effective field theory for quasicrystals and phasons dynamics

Phonons, phasons and atomic dynamics in quasicrystals - Chemical Society Reviews (RSC Publishing)

(3) The same holds where the authors mention disclinations and non-metricity. At least Kleinert book:

Gauge Fields in Condensed Matter (worldscientific.com)

must be referred.

(4) In line (61) the authors introduce a self-similar property of the points along the circles in fig.1. The reader deserves a bit more explanations here. Also, is it possible to quantify this better? For example, is it a fractal structure with a fractal dimension? How much is that fractal dimension and how it relates to the Φ of the projection?

(5)The caption of Fig.1 has some weird italic text. Please fix it.

(6) I am a bit confused by the discussion on the cylinder. In order to re-connect the disconnected lines, is a twist or a shear necessary? From fig.2 b1 is very clear is a shear strain. I am lost where the twist enters in this story. Maybe the authors could expand this a bit in the text and clarify it.

(7) In line 112 some quite non-common concepts appear. Either the authors explain them and the connection with the present work or they remove them.

(8) The discussion about M-theory, brane-worlds etc is not appropriate for a scientific paper. It is too speculative at this point and it just make the Reader feel this work is not serious. Please delete it and leave it for a future work in which you can clarify all these connections and not just throwing there buzzwords.

Author Response

Dear reviewer,

Thank you so much for your comments and suggestions - they are very helpful and inspiring. Please see attached documents for our response.

Thank you again! We really appreciate your help and your time!

Fang 

Reviewer 2 Report

This paper presents the periodicity in the perpendicular space of quasicrystals. This is of great fundamental interest for the community of quasicrystals.

Below a few comments

1- The introduction ma be improved by a more detailed bibliography

2- The explanation why it is a 2pi/phi periodicity is absent. (page 2 section 3)

3- Fig. 1 the color code for the circles is not explained. More generally, Fig. 1b is not clear. At least, the relationships between d_ij mentioned in Fig. 1a and some features of the circles in Fig. 1b should be mentioned.

4- Fig. 2c is shortly evoked in the paper. A detailed description of what is presented and what it implies would be welcome.

5- It is the same for Fig. 3. A more extensive explanation of the difference between a thinner and thicker window would greatly improve the paper.

6- The color code for atoms is not consistent in the whole paper: red in Fig. 2-3 and black/gray in Fig. 1

7- The x label in Fig. 1 only appears in Fig. 1 and seems not to be very useful in the paper.

Author Response

(The authors gave the same response as above.)

Round 2

Reviewer 2 Report

The authors have taken into account my comments. Modifications  have been done accordingly. I only have a single comment: it seems to me that no caption is provided for Fig. 2d.